# Shortness of breath in children at the emergency department: Variability in management in Europe

Dorine Borensztajn[1]☯*, Joany M. Zachariasse[1]‡, Susanne Greber-Platzer[2]‡, Claudio F. Alves[3]‡, Paulo Freitas[4]‡, Frank J. Smit[5]‡, Johan van der Lei[6]‡, Ewout W. Steyerberg[7]‡, Ian Maconochie[8]‡, Henriëtte A. Moll[1]☯

1 Department of General Paediatrics, Erasmus MC-Sophia Children's Hospital, University Medical Centre Rotterdam, Rotterdam, The Netherlands, 2 Department of Paediatrics and Adolescent Medicine, Medical University Vienna, Vienna, Austria, 3 Department of Paediatrics, Hospital Prof. Dr. Fernando da Fonseca, Lisbon, Portugal, 4 Intensive Care Unit, Hospital Prof. Dr. Fernando da Fonseca, Lisbon, Portugal, 5 Department of Paediatrics, Maasstad Hospital, Rotterdam, The Netherlands, 6 Department of Medical Informatics, Erasmus MC, University Medical Centre Rotterdam, Rotterdam, The Netherlands, 7 Department of Medical Statistics and Bioinformatics, Leiden University Medical Centre, Leiden, The Netherlands, 8 Department of Paediatric Accident and Emergency, Imperial College NHS Healthcare Trust, London, United Kingdom

☯ These authors contributed equally to this work.
‡ These authors also contributed equally to this work.
* d.borensztajn@erasmusmc.nl

**Data Availability Statement:** The minimal anonymized dataset has been uploaded to a public repository: https://doi.org/10.17026/dans-xhp-qbjg.

## Abstract

### Objective

Our aim was to describe variability in resource use and hospitalization in children presenting with shortness of breath to different European Emergency Departments (EDs) and to explore possible explanations for variability.

### Design

The TrIAGE project, a prospective observational study based on electronic health record data.

### Patients and setting

Consecutive paediatric emergency department visits for shortness of breath in five European hospitals in four countries (Austria, Netherlands, Portugal, United Kingdom) during a study period of 9–36 months (2012–2014).

### Main outcome measures

We assessed diversity between EDs regarding resource use (diagnostic tests, therapy) and hospital admission using multivariable logistic regression analyses adjusting for potential confounding variables.

**Funding:** No honorarium, grant, or other form of payment was given to anyone to produce the manuscript.

**Competing interests:** The authors declare they have no potential conflicts of interest to disclose.

**Abbreviations:** aOR, Adjusted Odds Ratio; CI, Confidence Interval; ED, Emergency Department; MTS, Manchester Triage System; OR, Odds Ratio; PICU, Paediatric Intensive Care Unit; TrIAGE, Triage Improvement Across General Emergency departments.

## Results

In total, 13,552 children were included. Of those, 7,379 were categorized as immediate/*very urgent*, ranging from 13–80% in the participating hospitals. Laboratory tests and X-rays were performed in 8–33% of the cases and 21–61% was treated with inhalation medication. Admission rates varied between 8–47% and PICU admission rates varied between 0.1–9%. Patient characteristics and markers of disease severity (age, sex, comorbidity, urgency, vital signs) could explain part of the observed variability in resource use and hospitalization. However, after adjusting for these characteristics, we still observed substantial variability between settings.

## Conclusion

European EDs differ substantially regarding the resource use and hospitalization in children with shortness of breath, even when adjusting for patient characteristics. Possible explanations for this variability might be unmeasured patient characteristics such as underlying disease, differences in guideline use and adherence or different local practice patterns.

## Introduction

Children with shortness of breath form a large part of children visiting the Emergency Department (ED) [1] and in many of these children some form of resource use such as diagnostic tests, treatment with inhalation medicine or hospital admission is initiated [1–5]. Although sometimes beneficial, these forms of resource use can also have negative effects on children, their families and the health care system, as they can be painful, cause anxiety, increase ED length of stay and ED crowding and increase health care costs [6]. Furthermore, higher resource use is not automatically associated with a higher quality of healthcare or improved outcomes [7].

Observing high variation may suggest that interventions are needed to improve the standardization of management of children with respiratory complaints visiting the ED.

Although several studies have been performed regarding resource use in children with shortness of breath, most studies were performed in the United States -which has a different health care system than most European countries—focused on a specific diagnosis which is often not available at arrival at the ED or on specific forms of resource use only, such as x-rays or antibiotic prescription rates [1–5].

The aim of our study was to describe patient characteristics in children presenting with shortness of breath in five different European EDs and to assess differences in resource use between EDs after adjusting for patient characteristics and markers of disease severity, in order to identify areas with a large variation in resource use.

## Methods

### Study design

This study was part of the TrIAGE project: *Triage Improvement Across General Emergency departments*, a prospective observational study consisting of a large cohort of children visiting the ED in 5 different hospitals in 4 European countries (UK, Austria, Portugal, the Netherlands) [8]. Participating study sites were diverse regarding type of hospital, catchment area,

number of ED visits, mixed adult-paediatric or paediatric ED and complexity of the patient population (S1 Table). Data collection consisted of routinely recorded, anonymized patient data, automatically extracted from electronic medical records. In order to increase completeness and accuracy of the data, we pre-specified a limited set of required variables. Data harmonization and quality checks were performed.

The study was approved by the medical ethical committees of the participating institutions: Medical Ethics Committee Erasmus MC (MEC-2013-567), Board of Directors Maasstad Ziekenhuis (L2013-103), Imperial College London Joint Research Compliance Office (14/WA/1051), Comissão de Ética para a Saúde do Hospital Prof. Dr. Fernando Fonseca EPE (Estudo Clínico TrIAGE—Parecer Favorável), Ethik Kommission Medizinische der Medizinischen Unversität Wien (EK Nr: 1405/2014). All waived the requirement for informed consent.

## Study population and setting

Participating hospitals were the Erasmus MC, Rotterdam, the Netherlands (January 2012 to December 2014); Maasstad Hospital, Rotterdam, the Netherlands (May 2014 to October 2015); St Mary's Hospital, London, United Kingdom (July 2014 to February 2015); Hospital Fernando da Fonseca, Lisbon, Portugal (March 2014 to February 2015); and General university Hospital, Vienna, Austria (January 2014 to December 2014).

Details regarding the participating hospitals are described in S1 Table.

We included all children until the age of 16 that consecutively attended the ED that presented with shortness of breath as a main complaint, defined as a Manchester triage system (MTS) flowchart of shortness of breath. The only exclusion criterion was missing data regarding presenting problem.

The MTS contains 52 flowcharts based on different presenting problems, including shortness of breath. Flowcharts contain discriminators that categorize patients into five categories of urgency, determining within how many minutes a patient should be seen by a physician [9].

Analysis included patient characteristics (age, sex, season of presentation) and markers of disease severity, such as comorbidity, increased work of breathing and vital signs: heart rate, respiratory rate, temperature, oxygen saturation and MTS urgency which was shown to be a reliable and valid reflection of urgency in children visiting the ED [10]. Vital signs were considered abnormal if they were outside the APLS reference ranges [11]. Fever was defined as a temperature of $\geq 38.0°C$. Patients with MTS triage category 1 and 2 (immediate and very urgent) were classified together as *very urgent*. Patients with triage category 3 (urgent) were classified as urgent and patients with triage category 4 and 5 (standard and non-urgent) were classified as non-urgent. To ensure consistency among hospitals, the urgency levels according to MTS version 3 including modifications for children with fever were modelled in all hospitals [12,13].

Comorbidity was documented in free text fields and was classified into complex comorbidity, noncomplex comorbidity, and no comorbidity based on the Paediatric Medical Complexity Algorithm. This algorithm uses ICD-9 codes to distinguish three groups of comorbidity. Patients were defined as having complex comorbidity if 2 or more body systems are affected, if they suffer from a progressive condition or malignancy [14].

Referral was categorized as self-referral, referral by a primary care physician, referral by ambulance or other (e.g. referral by specialist). To facilitate comparison by severity and distinguish patients with a more severe clinical presentations or ED course from those with a less severe presentation or ED course, we developed three clinical categories of severity. Children were classified as severe if they had triage category 1 (immediate) or 2 (very urgent), or had 2 or more abnormal vital signs or had severe work of breathing or received oxygen therapy or

received a life-saving intervention or were admitted to the PICU. Children were classified as non-severe if they had triage category 3, 4 or 5 and had no abnormal vital signs or increased work of breathing. All other children were classified as intermediate severity.

The main outcome measure were resource use and disposition. Resource use was categorized into diagnostics (laboratory tests and imaging) and therapy (inhalation or intravenous medication administered at the ED) at the ED.

Laboratory tests were defined as any blood test that was performed at the ED. Imaging was defined as any X-ray that was performed at the ED.

Inhalation medication was defined as any inhalation medication that was given at the ED. Intravenous medication was defined as either intravenous medication or intravenous fluid given at the ED, except in the Dutch teaching hospital, where only information on intravenous medication was available. Oral medication was not included in the analysis as this was not reliably available.

Immediate life-saving interventions were adapted from Lee et al. [15] and were categorized into the following categories: airway and breathing support, electrical therapy (e.g. cardioversion), emergent procedures, hemodynamic support and emergency medications (S2 Table).

Disposition was defined as discharge, admission and PICU admission after the ED visit.

## Data analysis

For vital signs, the first measurement at triage was used for analysis. Missing vital signs, capillary refill and level of consciousness were imputed using multiple imputation, a widely used technique for handling missing data, which has shown good performance in several studies [16,17].

The imputation model included age, sex, type of referral, MTS flowchart and urgency, time of arrival, vital sign values, work of breathing, capillary refill, level of consciousness, diagnostic tests, medication, oxygen therapy, life-saving interventions and disposition of all hospitals combined. This imputation process resulted in twenty-five databases on which statistical analysis were performed and pooled for a final result [18].

Imputation was performed by using the MICE imputation package in R, version 2.15.2.

We conducted descriptive analyses to explore differences between settings regarding patient characteristics, resource use and hospitalization. We used multivariable logistic regression models to assess the association between study site, resource use and hospitalization. Dependent variables were: hospital lab tests, x-ray's, inhalation medication, intravenous medication, hospital admission, and PICU admission.

The analyses were adjusted for age, sex, season, urgency level, abnormal vital signs and increased work of breathing, as these were considered potential confounding variables. All variables, including hospital, were included as categorical variables. Odds ratios were considered significant when the 95% confidence interval did not include 1. Analyses were performed using SPSS software (version 25.0).

## Results

### Patient characteristics and markers of disease severity

The total population consisted of 119,209 ED visits, of which 1,762 children (1.5%) were excluded due to a missing MTS flowchart. 13,552 (11.4%) children with shortness of breath were included for the analysis.

In all centers around half of all patients was below the age of 2 years. Patients presented most frequently in autumn or winter and least frequently in summer.

Children in the ED in Portugal were most often classified as immediate/very urgent (79.6%) while this was a minority in Austria (12.8%).

Most patients in the UK and Portugal were self-referred (81.9% and 95.9% respectively), while in the Dutch teaching hospital the majority was referred by a primary-care physician (72.0%).

Before imputation, vital signs were available with the following frequency: heart rate 93.7% (range between hospitals 91.0%-95.8%), respiratory rate 70.0% (range 15.2%-93.0%), temperature 94.9% (range 84.0%-99.7%) and oxygen saturation 97.0% (range 91.6%-99.8%).

The rate of tachypnoea varied between 3.9% in Austria and 49.2% in the UK. Fifty-nine percent of children were classified as severe (range between hospitals: 19.1–80.3%), 15% intermediate (range between hospitals 4.6–25.8%) and 26% non-severe (range between hospitals 13.7–55.1%). Detailed patient characteristics are shown in detail in Table 1.

## Practice variation between hospitals

Laboratory tests or X-rays were performed in 8–33% of all children. Inhalation medication was prescribed to 46% of all children and 38% of children younger than 1 year. After adjusting for patient characteristics and markers of severity the rate of laboratory tests was highest in the Austrian Hospital (adjusted OR 5.3, 95% CI 4.3–6.6) while X-rays were performed most often in the Portuguese hospital (adjusted odds ratio 8.8, 95% CI 7.0–11.1, Tables 2 and 3 and S3 Table).

After adjustment, inhalation medication was prescribed least often in the Austrian hospital (reference) and most often in the Portuguese hospital (adjusted OR 2.2, 95% CI 1.9–2.5). Intravenous medication was prescribed least often in the UK hospital (reference) and most often in the Dutch teaching hospital (adjusted OR 4.1 (95% CI 3.1–5.5) (S4 Table).

In total 2,362 children (17.4%) were admitted to a general ward. After adjustment, admission occurred most often in the Dutch tertiary hospital (adjusted OR 9.8, 95% 8.3–11.5) and least often in the Portuguese hospital (reference). Paediatric Intensive Care Unit (PICU) admission occurred most often in the Dutch tertiary hospital (adjusted OR 75.8, 95% CI 18.5–310.7, Tables 2 and 3 and S5 Table). This did not include interhospital transfers, as these children are admitted to the ICU directly.

Settings with a high rate of lab tests did not consistently have a high rate of imaging, therapy or hospital admission. Comorbidity could explain some but not all of the variance found between hospitals. Sub-analysis of the settings with ≥ 10% comorbidity (Vienna, London and the tertiary hospital in Rotterdam), showed higher resource use in children with comorbidity, especially in children with complex comorbidity (S6 Table). In the sub-analysis of children without comorbidity, the overall variability in resource use and hospitalization remained similar in the 5 EDs (S7 Table).

Furthermore, a sub-analysis of children with a past medical history of asthma, (three settings, N = 529) still shows considerable variation in resource use, including the use of inhalation medication or and lab tests (S8 Table).

To reduce the impact of the large number of missing values for respiratory rate in one setting, two additional analyses were performed. First, a sub-analysis excluding the setting with a large number of missing values for respiratory rate was performed, which showed similar variation between the remaining four EDs compared to the original analysis (S9 Table). Second, a complete case analysis for respiratory rate was performed, which showed similar variation compared to the original analysis (S10 Table).

**Table 1. Patient characteristics and markers of disease severity in percentage across EDs in Europe.**

| Number 13,552 overall | NL tertiary N = 1,558 N (%) | NL teaching N = 1,268 N (%) | UK N = 1,639 N (%) | PT N = 5,729 N (%) | AT N = 3,358 N (%) | P-value |
|---|---|---|---|---|---|---|
| Sex | | | | | | |
| Male | 961 (61.7) | 814 (64.2) | 1,028 (62.7) | 3,282 (57.3) | 1,904 (56.7) | <0.001 |
| Comorbidity | 848 (54.4) | (<10*) | 295 (18.0) | (<10*) | 367 (10.9) | <0.001 |
| Age | | | | | | <0.001 |
| 0–2 years | 798 (51.2) | 713 (56.2) | 736 (44.9) | 2,846 (49.7) | 1,572 (46.8) | |
| 2–5 years | 343 (22.0) | 346 (27.3) | 537 (32.8) | 1,517 (26.5) | 1,035 (30.8) | |
| ≥ 5 years | 417 (26.8) | 209 (16.5) | 366 (22.3) | 1,366 (23.8) | 751 (22.4) | |
| Referral | | | | | | <0.001 |
| Self | 419 (26.9) | 212 (16.7) | 1,336 (81.9) | 5,476 (95.9) | (>90*) | |
| Primary care | 405 (26.0) | 913 (72.0) | 36 (2.2) | 0 (0.0) | no data | |
| Emergency service | 139 (8.9) | 57 (4.5) | 89 (5.5) | 232 (4.1) | no data | |
| MTS urgency | | | | | | <0.001 |
| Immediate/very urgent | 753 (48.3) | 767 (60.5) | 867 (52.9) | 4,563 (79.6) | 429 (12.8) | |
| Urgent | 149 (9.6) | 348 (27.4) | 152 (9.3) | 117 (2.0) | 415 (12.4) | |
| Standard/non-urgent | 656 (42.1) | 153 (12.1) | 620 (37.8) | 1,049 (18.3) | 2,513 (74.8) | |
| Abnormal vital signs | | | | | | |
| Fever | 587 (37.7) | 455 (35.9) | 443 (27.0) | 645 (11.3) | 571 (17.0) | <0.001 |
| Oxygen saturation ≤ 94 | 220 (14.1) | 157 (12.4) | 132 (8.1) | 578 (10.1) | 162 (4.8) | <0.001 |
| Tachypnoea | 739 (47.7) | 593 (46.8) | 806 (49.2) | 1,822 (31.8) | 130 (3.9) | <0.001 |
| Tachycardia | 438 (28.1) | 446 (35.2) | 385 (23.5) | 1,345 (23.5) | 481 (14.3) | <0.001 |
| Increased work of breathing | 995 (63.9) | 677 (53.4) | 913 (55.7) | 4,393 (76.7) | 557 (16.6) | <0.001 |
| Severity classification | | | | | | <0.001 |
| Severe | 959 (61.6) | 870 (68.6) | 968 (59.1) | 4599 (80.3) | 641 (19.1) | |
| Intermediate | 385 (24.7) | 218 (17.2) | 281 (17.1) | 266 (4.6) | 866 (25.8) | |
| Non-severe | 214 (13.7) | 180 (14.2) | 390 (23.8) | 864 (15.1) | 1851 (55.1) | |

NL teaching = Maasstad Hospital, Rotterdam, the Netherlands; NL tertiary = Erasmus MC, Rotterdam, the Netherlands; UK = St Mary's Hospital, London, United Kingdom; PT = Hospital Fernando da Fonseca, Lisbon, Portugal; AT = General Hospital, Vienna, Austria.

* * overall data for this setting, not available for each individual patient.

## Practice variation by age groups

Table 2 describes variation in resource use for different age groups. In general, EDs that had high rates of resources in all patients, showed comparable high rates in the age group below or above 1 year. There was no clear pattern between age group and resource use, e.g. in some EDs resource use was higher in younger children while in other settings resource use was higher in older children; no increased uniformity in resource use between EDs was observed when looking separately at young children or children above 1 year old (Tables 2 and 3 and S3–S5 Tables).

## Practice variation by severity

In children with a severe presentation the rate of laboratory tests, x-rays and iv medication was higher at three EDs, while in the other two settings there was no significant difference between severity groups. There was no relationship between severity and the prescription of inhalation medication at any setting. In 4 out of 5 EDs, children with a severe presentation were admitted more often.

**Table 2. Heatmap for different ages with odds ratios of resource use, corrected for patient characteristics#.**

|  | NL tertiary | NL teaching | UK | PT | AT |
|---|---|---|---|---|---|
|  | aOR | aOR | aOR | aOR | aOR |
| Blood tests all children | 3.9* | 1.2** | + | 1.4* | 5.3* |
| < 1 year | 3.4* | 1.2** | + | 1.8* | 5.8* |
| > 1 year | 4.1* | 1.3** | + | 1.2** | 5.1* |
| X-rays all children | 5.0* | + | 2.3* | 8.8* | 4.1* |
| < 1 year | 11.6* | + | 5.6* | 17.7* | 11.2* |
| > 1 year | 4.0* | + | 1.9* | 7.5* | 3.3* |
| Inhalation medication all children | 1.1** | 1.6* | 1.7* | 2.2* | + |
| < 1 year | 1.2** | 1.8* | + | 3.0* | 1.8* |
| > 1 year | 1.3* | 1.8* | 2.4* | 2.4* | + |
| Intravenous medication all children | 2.0* | 4.1* | + | 1.3** | 1.1** |
| < 1 year | 2.5* | 3.7* | + | 1.3** | 1.6** |
| > 1 year | 1.7* | 4.4* | 1.0** | 1.2** | + |
| General admission all children | 9.8* | 7.6* | 4.0* | + | 2.2* |
| < 1 year | 6.6* | 4.6* | 1.4* | + | 1.4* |
| > 1 year | 12.6* | 11.1* | 6.8* | + | 2.9* |
| ICU admission all children | 75.8* | 1.7** | 2.6** | 8.5* | + |
| < 1 year | 78.0* | + | 1.2** | 15.9* | 1.8** |
| > 1 year | 117.7* | 3.4** | 4.9** | 7.1** | + |

#Associations are determined by multivariable logistic regression models. Model adjusted for sex, age, season, triage urgency, fever, tachycardia, tachypnoea, low oxygen saturation and increased work of breathing.

+ reference.

* P-value <0.01.

** not significant.

NL teaching = Maasstad Hospital, Rotterdam, the Netherlands; NL tertiary = Erasmus MC, Rotterdam, the Netherlands; UK = St Mary's Hospital, London, United Kingdom; PT = Hospital Fernando da Fonseca, Lisbon, Portugal; AT = General Hospital, Vienna, Austria.

In general, EDs that had high or low rates of resources, had similar trends in the severe as well as the non-severe group (Table 3 and S3–S5 Tables). When performing a separate analysis after removing patients with triage urgency immediate (the highest acuity group), similar trends with high variability in all types of resource use were seen (S11 Table).

## Discussion

### Statement of principal findings and findings in relation to the literature

Our results show marked variability in the resource use and hospitalization in children presenting to the ED with shortness of breath. Marked variability between EDs remained after adjusting for patient characteristics and markers of disease severity.

Subgroup analyses by age groups still showed considerable variation between hospitals.

In general, EDs that had high or low rates of specific types of resource use, showed similar patterns in different age or severity groups.

Overall, variation was lower in the severe group than in the non-severe group and showed considerable variation between settings; variation in the non-severe group was highest in diagnostic tests and therapy.

Regarding specific forms of resources, variation was most pronounced in the performance of x-rays, especially in children below 1 year of age.

**Table 3. Heatmap for patients with different severity with odds ratios of resource use, corrected for patient characteristics[#].**

|  | NL tertiary | NL teaching | UK | PT | AT |
|---|---|---|---|---|---|
|  | aOR | aOR | aOR | aOR | aOR |
| Blood tests all children | 3.9 | 1.2** | + | 1.4 | 5.3 |
| severe | 3.6 | + | 1.0** | 1.3** | 5.6 |
| non severe | 14.3 | 4.7 | + | 4.3 | 11.7 |
| X-rays all children | 5.0 | + | 2.3 | 8.8 | 4.1 |
| severe | 4.7 | + | 2.3 | 8.0 | 4.3 |
| non severe | 9.1 | + | 2.3** | 21.0 | 7.3 |
| Inhalation medication all children | 1.1** | 1.6 | 1.7 | 2.2 | + |
| severe | + | 1.5 | 1.7 | 2.0 | 1.0** |
| non severe | + | 2.9 | 2.0 | 4.6 | 1.8 |
| Intravenous medication all children | 2.0 | 4.1 | + | 1.3** | 1.1** |
| severe | 1.8 | 3.0 | 1.0** | + | 1.6 |
| non severe | 9.9 | 65.8 | + | 15.1 | 4.3** |
| General admission all children | 9.8 | 7.6 | 4.0 | + | 2.2 |
| severe | 10.0 | 6.9 | 4.1 | + | 2.4 |
| non severe | 10.7 | 12.6 | 1.9** | + | 1.6** |
| ICU admission all children | 75.8 | 1.7** | 2.6** | 8.5 | + |
| severe | 52.2 | 1.4** | 2.0** | 6.2 | + |
| non severe | n.a. | n.a. | n.a. | n.a. | n.a. |

[#]Associations are determined by multivariable logistic regression models. Model adjusted for sex, age, season, triage urgency, fever, tachycardia, tachypnoea, low oxygen saturation and increased work of breathing.

[+] reference.

[*] P-value <0.01.

[**] not significant.

NL teaching = Maasstad Hospital, Rotterdam, the Netherlands; NL tertiary = Erasmus MC, Rotterdam, the Netherlands; UK = St Mary's Hospital, London, United Kingdom; PT = Hospital Fernando da Fonseca, Lisbon, Portugal; AT = General Hospital, Vienna, Austria.

n.a. = not applicable.

Furthermore, considerable variation in hospital admission was seen in patients of all ages and both in the severe and non-severe group.

Comparable to our results, previous studies have described high variability in children presenting to the ED with respiratory problems such as bronchiolitis, asthma, croup and pneumonia, even in the presence of guidelines [1–5]. Moreover, in our study this variability remained after correcting for several patient characteristics and markers of disease severity.

Factors which might explain this variation include seasonal trends, availability of immunization, availability of primary care, type of ED or physician background [1,2,5,19], parental demand [20], availability of point-of-care tests [21,22], and variability, availability and adherence to guideline content [23–25].

A recent study on guideline availability found that the use of inhalation medication in the management of bronchiolitis had increased despite national guidance to withhold inhalation medication [25].

Specific seasonal [26], and geographical [27] trends are known to influence the incidence of respiratory tract infections and thus consequently can influence resource use. Furthermore, immunization and immunization rates differ per country [28,29]. For example, while immunization against Haemophilus Influenzae Type B and pneumococcal disease is offered routinely in all participating countries, influenza immunization is offered routinely to children in the

UK only [28]. Although we did not collect detailed data regarding (viral) diagnosis, as our data collection consisted of at least 12 months and including all seasons in all settings but one and we included season in our analysis, this did not explain the marked differences we found between EDs.

Primary care availability and mode of referral can have an impact on the ED patient case mix and consequently on resource use and hospitalization [30,31].

Our study showed marked variability in self-referral rates between different settings as in some settings a primary care physician was not available, or was bypassed, especially during out-of-office hours. Although we did not adjust for mode of referral in the whole analysis as detailed referral information was available in four out of five settings, sub-analysis in these four settings still showed large variation after adjusting for mode of referral (S12 Table). Furthermore, we did adjust for patient case mix by adjusting for markers of disease severity and thus indirectly adjusted for mode of referral, as this influences patient case mix.

Previous research showed that type of ED (dedicated paediatric or mixed) and physician background [1,2,5,19] can influence resource use and hospitalization. As in our dataset most settings were a dedicated paediatric ED and in all but one setting children were seen exclusively by a paediatrician, it is not likely that this factor alone explains the variation we found in our study.

We used standardized routine data for our study. Using large datasets of routinely obtained observational data ("big data") is gaining in popularity [32], as it poses many benefits, such as presenting large amounts of patient data [32] with opportunities for generating new insights and analyzing complex correlations between patient factors and outcomes.

However, one must realize that data from different settings are not always comparable and other factors than patient-factors alone, such as local factors can influence resource use and hospitalization. Therefore, combining these datasets and interpreting results should be done with caution and with an eye on local factors. Even outcome measures that are considered to be objective and comparable at first glance, such as hospital and PICU admission, can be influenced by local protocols and thus can reflect other factors than disease severity alone.

## Implications for clinical practice and research

Variability in management reflects high and potentially avoidable resource use, which increases health care costs, ED crowding, adverse events and reduces quality of care [3,33].

Reduction of this variability is a good starting point to decrease healthcare costs and improve quality of care [33].

The use of clinical practice guidelines can potentially reduce variability and improve quality of care [3], however, the mere availability of guidelines does not automatically entail guideline adherence [1,34–36]. Several studies have shown that interventions to improve guideline adherence reduce resource use without negatively affecting other outcomes such as hospital admission rates [23,37–40]. Furthermore, other interventions, such as bedside viral tests or a bedside CRP test, can positively impact resources use, such, x-rays, antibiotic use and hospital admission [22].

Our data identifies a large group of children that can benefit from adherence to guidelines and bedside tests.

## Strengths and limitations

To our knowledge, this is the first study that describes variability in the management of children with shortness of breath at European EDs. The main strength of our study includes the large number of patients and that data were collected from 5 settings in 4 European countries,

which increases the generalizability of the results. Furthermore, university as well as non-university hospitals participated in this study.

The enrolment period varied between the different hospitals. However, each hospital was followed for at least 8 months and in all hospitals a winter season was included to increase the comparability between settings. In one setting, only information on intravenous medication was available and not intravenous fluid, but even with this information lacking, this setting still showed the highest use of intravenous therapy. For all other variables, uniform definitions were used.

Our data does not contain information regarding diversity in antibiotic use. Antibiotic use, however, has been extensively studied previously [41] and only a minority of acute respiratory problems in children are of bacterial origin [42]. In this study we focused on initial diagnostic tests, inhaled medication, iv medication and hospitalization.

Although we adjusted for patient characteristics and markers of disease severity, we cannot exclude that unmeasured aspects of patient or setting characteristics, such as past medical history, tests or interventions performed prior to arrival at the ED, immunization status, availability of primary care or physician experience and background, could still be of influence.

We used presenting symptom as the main inclusion criterion and not final diagnosis, which is usually not available in the early phase of the diagnostic process. This resulted in a heterogeneous study population of children with different final diagnoses, such as upper respiratory tract infections, viral and bacterial lower respiratory tract infections, viral induced wheezing and asthma. Although our study included children with these different final diagnoses, and in part management depends on the final diagnosis, there is also considerable overlap in the management of these diagnoses.

At the ED final diagnoses are missing in many cases and diagnostics and interventions are based on a working diagnosis. Discrimination between different cause of respiratory illness can be difficult. Participating countries' national guidelines regarding the main final diagnoses of children visiting the ED with shortness of breath, such as asthma, bronchiolitis and pneumonia, clearly state that diagnostic tests such as blood tests or x-rays have no place in the assessment of children with any of these diagnoses, except in atypical or severe cases (e.g. children that require PICU admission) [43–46], except the National Portuguese guidelines [47] which advice to perform lab tests in all children with suspected bacterial disease and X-rays in children with bronchiolitis with suspected bacterial infection. Although this might explain the high rate of X-rays that were seen in the Portuguese setting, our data still showed large variation in resource use and hospitalization in the severe as well as the non-severe group of children with respiratory symptoms that cannot be explained by guideline content. Furthermore, inhalation medication is not recommended in children with respiratory complaints and wheezing below the age of 1 year, as these complaints are usually caused by bronchiolitis, and there is no evidence of a positive effect of inhalation medication in this age group. However, we still found high use and high variation of inhalation medication in this specific age group [48]. Lastly, hospital admission in general is guided by disease severity and not by final diagnosis [43–47].

Finally, our results should be interpreted in light of the limitations of using standardized routinely collected data and missing data. In our study, there was a number of missing vital signs, especially respiratory rate. However, this reflects actual routine care at many EDs [49] and is one of the limitations of using routine data. For example, the multicenter study of van de Maat et al. showed that complete vital sign measurement (temperature, heart rate, respiratory rate and capillary refill) occurred in only half of pediatric ED visits and respiratory rate was the vital sign missing most frequently [49]. In our study, the median number of vital sign measurement and other clinical variables were large enough to perform multiple imputation

of the missing values. When using routine data for research purposes, multiple imputation is considered a useful strategy for handling missing data and superior to other methods; this statement is supported by several simulation studies, including simulation studies showing unbiased results even with large proportions of missing data, provided sufficient auxiliary information was available [16–18,50,51]. Furthermore, we performed two sub-analyses, excluding the setting with the largest number of missing values for respiratory rate, and a complete case analysis for respiratory rate and found similar variation between ED's in comparison to the results of the original imputed dataset.

## Conclusion

Our data show that large variability in resource use and hospitalization exist in children with respiratory complaints visiting European EDs and that variability remains high even after adjusting for patient characteristics and disease severity. Possible explanations could be differences in guideline content or adherence or differences in local practice patterns. Our data show that given the high variability we found, there is a large opportunity for improvement of management and reduction of health care costs in this common clinical problem.

"What is already known on this topic"

- Resource use and hospitalization rates at the ED are widely variable.

- Resource use is influenced by patient level factors and hospital level factors

  "What this study adds"

- Large variability exist in European EDs regarding the use of diagnostic tests, initiated therapy and hospitalization in children with shortness of breath.

- Variability in management between settings cannot be explained by severity of symptoms only.

- Children with shortness of breath form a large patient group with room for improvement regarding uniform management and reduction of resource use.

## Supporting information

**S1 Table. Description of the different study sites.**
(PDF)

**S2 Table. Immediate life-saving interventions.**
(PDF)

**S3 Table. Differences in diagnostic tests between EDs.**
(PDF)

**S4 Table. Differences in medication between EDs.**
(PDF)

**S5 Table. Differences in hospital admission between EDs.**
(PDF)

**S6 Table. Differences in resource use in children with and without comorbidity.**
(PDF)

**S7 Table. Heatmap with odds ratios of resource use, excluding patients with comorbidity.**
(PDF)

**S8 Table. Heatmap with odds ratios of resource use for children with a past medical history of asthma.**
(PDF)

**S9 Table. Heatmap with odds ratios of resource use, excluding patients from the ED in Austria.**
(PDF)

**S10 Table. Heatmap with odds ratios of resource use, complete case analysis for respiratory rate.**
(PDF)

**S11 Table. Heatmap with odds ratios of resource use, excluding patients with triage urgency "immediate".**
(PDF)

**S12 Table. Heatmap with odds ratios of resource use, including correction for referral.**
(PDF)

## Author Contributions

**Conceptualization:** Dorine Borensztajn, Johan van der Lei, Henriëtte A. Moll.

**Data curation:** Dorine Borensztajn, Joany M. Zachariasse, Susanne Greber-Platzer, Claudio F. Alves, Paulo Freitas, Frank J. Smit, Johan van der Lei, Ian Maconochie, Henriëtte A. Moll.

**Formal analysis:** Joany M. Zachariasse.

**Investigation:** Dorine Borensztajn, Joany M. Zachariasse, Susanne Greber-Platzer, Claudio F. Alves, Paulo Freitas, Frank J. Smit, Johan van der Lei, Ian Maconochie, Henriëtte A. Moll.

**Methodology:** Ewout W. Steyerberg, Henriëtte A. Moll.

**Project administration:** Henriëtte A. Moll.

**Supervision:** Henriëtte A. Moll.

**Writing – original draft:** Dorine Borensztajn.

**Writing – review & editing:** Dorine Borensztajn, Joany M. Zachariasse, Susanne Greber-Platzer, Claudio F. Alves, Paulo Freitas, Frank J. Smit, Johan van der Lei, Ewout W. Steyerberg, Ian Maconochie, Henriëtte A. Moll.

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
