## [Decision Letter · Decision Letter 0]

2 Oct 2020

PONE-D-20-19274

Shortness of breath in children at the emergency department: variability in management in Europe.

PLOS ONE

Dear Dr. Borensztajn,

Thank you for submitting your manuscript to PLOS ONE. After careful consideration, we feel that it has merit but does not fully meet PLOS ONE’s publication criteria as it currently stands. Therefore, we invite you to submit a revised version of the manuscript that addresses the points raised during the review process.

We look forward to receiving your revised manuscript.

Kind regards,

Brenda M. Morrow, PhD

Academic Editor

PLOS ONE

Journal Requirements:

"The study was approved by the institutional review boards of all participating institutions and the need for informed consent was waived.".   

i) Please amend your current ethics statement to include the full name of the ethics committee/institutional review board(s) that approved your specific study.

ii) Once you have amended this/these statement(s) in the Methods section of the manuscript, please add the same text to the “Ethics Statement” field of the submission form (via “Edit Submission”).

Reviewers' comments:

Reviewer's Responses to Questions

**Comments to the Author**

1. Is the manuscript technically sound, and do the data support the conclusions?

Reviewer #1: Partly

Reviewer #2: Partly

2. Has the statistical analysis been performed appropriately and rigorously? 

Reviewer #1: I Don't Know

Reviewer #2: I Don't Know

3. Have the authors made all data underlying the findings in their manuscript fully available?

Reviewer #1: Yes

Reviewer #2: Yes

4. Is the manuscript presented in an intelligible fashion and written in standard English?

Reviewer #1: Yes

Reviewer #2: Yes

5. Review Comments to the Author

Reviewer #1: This article examines the variation in management of children presenting to the emergency departments across multiple institutions. This variation could represent opportunities for standardization of management to improve patient care.

Major:

-Imputation used for missing values. RR range of completion reported 15-93% which concerns me for this to be used as one of the main variables used for an illness severity marker.

-Discussion organization can be improved, current form seems to be more of a list of findings, would be helpful to add how this compares to other studies and applying the findings to clinical practice. I would focus more on why variation matters (cost, outcomes, etc) and that guidelines have been shown to improve this.

Minor

-How were comorbidities obtained? Problem list versus billing data?

- Knowing the specific respiratory past medical history would be helpful in interpreting data, especially the use of inhalation medications (i.e. asthma, BPD)

-How was increased work of breathing measured? Subjective by provider versus scoring system, if the latter is it a validated tool?

-Were vital signs recorded just initial or ongoing?

-What is electrical therapy? (line 181)

-Results: Good sample size overall, but almost half of encounters were from same hospital

-Statement majority had abnormal respiratory rate, but <50% at each site reported to be tachypnic in table 1

-abstract “categorized as very urgent ranging from 13-80%" (but in results you give 12.1 and 77% as the range)

- Alternate between commas and periods used as decimal points in text and tables

- Line 264-265 add p-value for sub-analysis

-Limitations that should be highlighted: missing factors noted to impact resource use in other studies not included in this study, unable to account for relevant pmh specific to management approach, unknown what testing/management prior to arrival for referred patients

-appendix 1 include p-values comparing sites. What does the estimation annotation mean?

Reviewer #2: This is an interesting study focused on a different way of approaching the evaluation of variability in healthcare settings. This focuses on the chief complaint of shortness of breath as opposed to diagnosis-based assessment of variability. However, they did perform statistical analysis to try to adjust for the diversity of patient population both in acuity and other possible confounding factors. The triage process was standardized throughout hospitals.

Consider removing Triage level 1 acuity (highest acuity) as this population likely should not be a population that decreases variation, in fact most patients in this category likely get a large amount of resource utilization which is likely appropriate.

With the analysis did you cluster by hospital center?

I have some concerns that the respiratory rate was missing most frequently (70% of the time, but down to 15% at its worst) in a manuscript that is addressing variability in patient presenting with shortness of breath. Can you explain this?

I think my largest concern is related to whether or not your dataset has data related to patient diagnosis to be able to adjust for differences related to a patient resource utilizations with pneumonia versus asthma. As those two presentations although may have the same chief complaint, they likely require two different workups and are based on an provider assessment.

You should consider that the tertiary center may have increased use of the ICU due to transfers from outside hospitals and have higher acuity patients. Did you account for interhospital transfers?

6. PLOS authors have the option to publish the peer review history of their article (what does this mean?). If published, this will include your full peer review and any attached files.

Reviewer #1: **Yes: **Aleisha M Nabower

Reviewer #2: No

---

## [Author Response · Author response to Decision Letter 0]

30 Nov 2020

Dear Ms. Morrow, dear reviewers

Thank you for your time, effort and expertise in reviewing our manuscript and taking it under consideration for publication.

We have adapted the manuscript according to the comments and we think that the manuscript improved after these important comments.

Please find our response uploaded as an attachment. 

kind regards

Dorine Borensztajn

---

## [Decision Letter · Decision Letter 1]

2 Feb 2021

PONE-D-20-19274R1

Shortness of breath in children at the emergency department: variability in management in Europe.

PLOS ONE

Dear Dr. Borensztajn,

Thank you for submitting your manuscript to PLOS ONE. After careful consideration, we feel that it has merit but does not fully meet PLOS ONE’s publication criteria as it currently stands. Therefore, we invite you to submit a revised version of the manuscript that addresses the points raised during the review process.

We look forward to receiving your revised manuscript.

Kind regards,

Brenda M. Morrow, PhD

Academic Editor

PLOS ONE

Reviewers' comments:

Reviewer's Responses to Questions

**Comments to the Author**

1. If the authors have adequately addressed your comments raised in a previous round of review and you feel that this manuscript is now acceptable for publication, you may indicate that here to bypass the “Comments to the Author” section, enter your conflict of interest statement in the “Confidential to Editor” section, and submit your "Accept" recommendation.

Reviewer #2: (No Response)

2. Is the manuscript technically sound, and do the data support the conclusions?

Reviewer #2: Partly

3. Has the statistical analysis been performed appropriately and rigorously? 

Reviewer #2: I Don't Know

4. Have the authors made all data underlying the findings in their manuscript fully available?

Reviewer #2: Yes

5. Is the manuscript presented in an intelligible fashion and written in standard English?

Reviewer #2: Yes

6. Review Comments to the Author

Reviewer #2: I appreciate the authors addressing many of the questions that I raised. My one major concern still surrounds the following:

1) Use of the multiple imputation methodology in a manuscript describing variability in approach to shortness of breath with the vital sign that is missing most being the respiratory rate. Multiple imputation seems to be a helpful tool but I worry when it is replacing missing data within a primary variable (respiratory rate) related to the manuscripts main outcome. While I recognize that this data may not exist I worry about making conclusions based on this methodology. You could consider removing the hospital that has only 15% of the respiratory rate vital sign present with the hopes that you increase the median enough to make this more valid and hopefully supporting an accurate conclusion.

7. PLOS authors have the option to publish the peer review history of their article (what does this mean?). If published, this will include your full peer review and any attached files.

Reviewer #2: No

---

## [Author Response · Author response to Decision Letter 1]

3 Apr 2021

Dear editor, dear reviewers

Thank you for your time, effort and expertise in reviewing our manuscript and taking it under consideration for publication.

We have adapted the manuscript according to the comments and we think that the manuscript improved after these important comments.

Please find our response attached as a separate file.

---

## [Decision Letter · Decision Letter 2]

20 Apr 2021

Shortness of breath in children at the emergency department: variability in management in Europe.

PONE-D-20-19274R2

Dear Dr. Borensztajn,

We’re pleased to inform you that your manuscript has been judged scientifically suitable for publication and will be formally accepted for publication once it meets all outstanding technical requirements.

Kind regards,

Brenda M. Morrow, PhD

Academic Editor

PLOS ONE

Additional Editor Comments (optional):

Reviewers' comments:

Reviewer's Responses to Questions

**Comments to the Author**

1. If the authors have adequately addressed your comments raised in a previous round of review and you feel that this manuscript is now acceptable for publication, you may indicate that here to bypass the “Comments to the Author” section, enter your conflict of interest statement in the “Confidential to Editor” section, and submit your "Accept" recommendation.

Reviewer #2: All comments have been addressed

2. Is the manuscript technically sound, and do the data support the conclusions?

Reviewer #2: Yes

3. Has the statistical analysis been performed appropriately and rigorously? 

Reviewer #2: I Don't Know

4. Have the authors made all data underlying the findings in their manuscript fully available?

Reviewer #2: Yes

5. Is the manuscript presented in an intelligible fashion and written in standard English?

Reviewer #2: Yes

6. Review Comments to the Author

Reviewer #2: I feel that the authors have adequately addressed my concerns that were raised in the previous reviews. Thank you.

7. PLOS authors have the option to publish the peer review history of their article (what does this mean?). If published, this will include your full peer review and any attached files.

Reviewer #2: No

---

## [Editor Report · Acceptance letter]

23 Apr 2021

PONE-D-20-19274R2 

Shortness of breath in children at the emergency department: variability in management in Europe. 

Dear Dr. Borensztajn:

I'm pleased to inform you that your manuscript has been deemed suitable for publication in PLOS ONE. Congratulations! Your manuscript is now with our production department. 

Kind regards, 

on behalf of

Professor Brenda M. Morrow 

Academic Editor

PLOS ONE